# Protocol for a prospective double-blind, randomised, placebo-controlled feasibility trial of octreotide infusion during liver transplantation

Jeremy Fabes  ,[1,2] Gareth Ambler,[3] Bina Shah,[4] Norman R Williams,[4] Daniel Martin,[1] Brian R Davidson,[4] Michael Spiro[4]

[1]Peninsula Medical School, University of Plymouth, Plymouth, UK
[2]Department of Anaesthesia, Royal Free London NHS Foundation Trust, London, UK
[3]Department of Statistical Science, University College London, London, UK
[4]Division of Surgery & Interventional Science, University College London, London, UK

**Correspondence to**
Dr Jeremy Fabes;
jeremy.fabes@nhs.net

## ABSTRACT

**Introduction** Liver transplantation is a complex operation that can provide significant improvements in quality of life and survival to the recipients. However, serious complications are common and include major haemorrhage, hypotension and renal failure. Blood transfusion and the development of acute kidney injury lead to both short-term and long-term poor patient outcomes, including an increased risk of death, graft failure, length of stay and reduced quality of life. Octreotide may reduce the incidence of renal dysfunction, perioperative haemorrhage and enhance intraoperative blood pressure. However, octreotide does have risks, including resistant bradycardia, hyperglycaemia and hypoglycaemia and QT prolongation. Hence, a randomised controlled trial of octreotide during liver transplantation is needed to determine the cost-efficacy and safety of its use; this study represents a feasibility study prior to this trial.

**Methods and analysis** We describe a multicentre, double-blind, randomised, placebo-controlled feasibility study of continuous infusion of octreotide during liver transplantation surgery. We will recruit 30 adult patients at two liver transplant centres. A blinded infusion during surgery will be administered in a 2:1 ratio of octreotide:placebo. The primary outcomes will determine the feasibility of this study design. These include the recruitment ratio, correct administration of blinded study intervention, adverse event rates, patient and clinician enrolment refusal and completion of data collection. Secondary outcome measures of efficacy and safety will help shape future trials by assessing potential primary outcome measures and monitoring safety end points. No formal statistical tests are planned. This manuscript represents study protocol number 1.3, dated 2 June 2021.

**Ethics and dissemination** This study has received Research Ethics Committee approval. The main study outcomes will be submitted to an open-access journal.

**Trial sponsor** The Joint Research Office, University College London, UK.
Neither the sponsor nor the funder have any role in study design, collection, management, analysis and interpretation of data, writing of the study report or the decision to submit the report for publication.

**Trial registration** The study is registered with ClinicalTrials.gov (NCT04941911) with recruitment due to

## Strengths and limitations of this study

► This study is prospective, randomised and designed to minimise the risk of bias throughout.
► We have prioritised patient input throughout the development and design of this study.
► A small sample size has been selected to demonstrate feasibility end points as a precursor to a larger efficacy and safety trial.
► The secondary end points will provide clinically relevant data on which to design this cost effectiveness study.

start in August 2021 with anticipated completion in July 2022.
**Clinical trials unit** Surgical and Interventional Group, Division of Surgery & Interventional Science, University College London.

## INTRODUCTION

Liver transplantation is a potentially life transforming intervention with excellent outcomes (1-year and 5-year survival of 94% and 84%, respectively).[1] It is, however, a major operation in patients with multiple comorbidities and is associated with a 3% perioperative mortality rate and 70% complication rate. Intraoperative haemorrhage is serious and commonly occurs during mobilisation of the diseased liver, exacerbated by coagulopathy and abnormal vasculature due to portal hypertension. Bleeding can cause hypotension, reduced perfusion of the transplanted graft and increased incidence of acute kidney injury (AKI). Blood loss and transfusion are associated with increased morbidity, mortality and cost of transplantation and can have a long-term detrimental effect on patient wellbeing and quality of life.[2–4]

Octreotide is a synthetic analogue of somatostatin with an excellent safety profile[5] that may improve renal outcomes, reduce

perioperative bleeding and increase exogenous vasopressor sensitivity during liver transplantation. AKI following liver transplantation occurs in 25%–60% of patients with normal renal function prior to surgery, with half of these patients requiring renal replacement therapy (RRT) after surgery.[6 7] The need for post-transplant RRT increases the length of hospital stay and the risk of early allograft dysfunction, chronic kidney disease (CKD) and mortality.[8 9] Risk factors for renal dysfunction following transplantation include intraoperative hypotension, lower quality donor grafts, hepatic ischaemia-reperfusion injury, blood transfusion and preoperative renal disease.[10 11] Octreotide has been shown to enhance renal blood flow, glomerular filtration and urine output in decompensated liver cirrhosis,[12] hepatorenal syndrome[13] and may also reduce the incidence of post-transplant renal injury.[14 15]

Major haemorrhage and transfusion during and after liver transplantation reduce patient and graft survival as well as increasing length of stay and the need for RRT.[2–4 16–18] Haemorrhage risk during liver transplantation is multifactorial, including cirrhotic coagulopathy, portal hypertension and hyperfibrinolysis. Bleeding is driven by the presence of widespread dilated collaterals and a hyperdynamic splanchnic circulation. Octreotide causes splanchnic vasoconstriction thereby reducing splanchnic blood flow, collateralisation and portal venous pressures which may reduce surgical blood loss.[19] This correction of the abnormal blood distribution and venous pooling increases systemic blood pressure[20] and, when combined with a vasopressor, correlates with improved renal function.[21] Furthermore, increased systemic pressures and vasopressor sensitivity help preserve renal function and reduce myocardial injury.[10 22 23]

### Previous clinical studies of octreotide in liver transplantation

A retrospective cohort study of octreotide infusion during liver transplantation demonstrated a reduction in transfusion requirement by 2.3 units of packed red blood cells but this difference was not statistically significant.[24] A randomised controlled trial (RCT) of octreotide infusion during liver transplantation found an enhancement of norepinephrine effect on blood pressure with a mean increase of 8 mm Hg.[15]

### Risks

Octreotide is not currently licensed for use in liver transplantation. Common risks of octreotide include changes in blood sugar levels, heart rate and rhythm. The side effect profile of octreotide during liver transplantation is unknown. The studies to date did not compare safety or adverse events between the octreotide and placebo arms,[15] have not been fully published[25] or did not assess adverse events.[24]

### Rationale for this study

The background literature for the use of octreotide in liver transplantation is inconclusive and the current use in this setting is clinician and institution dependent. While there is reasonable evidence to support its use in this setting, octreotide has harmful effects which need to be considered against its efficacy and cost. Furthermore, patients may be unwilling to have non-essential medication and clinicians may not support its use in all transplant settings. In addition to determining the feasibility of a multisite drug trial in liver transplantation, secondary end points will provide clinically relevant data on which to design a subsequent cost-effectiveness study.

### Hypothesis

It is feasible to recruit and retain patients in a randomised study of continuous infusion of octreotide during liver transplantation surgery.

## METHODS AND ANALYSIS
### Trial design

The purpose of this study is to determine the feasibility of recruiting patients to a double-blind, randomised, placebo-controlled trial of intravenous octreotide during liver transplantation. We will recruit 30 adult patients at two liver transplant centres; The Royal Free London NHS Foundation Trust (Royal Free Hospital, RFH; National Health Service, NHS), and University Hospitals Birmingham NHS Foundation Trust (University Hospital Birmingham). A blinded infusion will be administered during transplantation surgery. Patients will be allocated in a 2:1 ratio to octreotide or placebo. All members of the research, anaesthetic, surgical, intensive care and nursing teams will be blinded to patient allocation; this includes all outcome assessors and data analysts. Patients will be blinded to their allocation and will have the option to find out which intervention they received after the study is complete. Unblinding will be carried out once data cleaning and analysis has been completed.

This protocol conforms to the Standard Protocol Items: Recommendations for Interventional Trials.[26] The RCT will be reported in line with the Consolidated Standards of Reporting Trials statement.[27] This trial is non-commercial and does not intend to generate data for drug licensing. The study is registered with ClinicalTrials.gov (NCT04941911) with recruitment due to start in August 2021 with anticipated completion in July 2022. WHO Trial Registration Data Set in table 1.

### Eligibility
#### Inclusion criteria
1. Adults aged 18 years and over accepted for liver transplant and on the transplant waiting list.
2. Undergoing liver transplantation.
3. Recipient of a whole or partial liver graft from a cardiac or brain dead donor.

#### Exclusion criteria
1. Lacking capacity to provide consent (according to the Mental Capacity Act, 2005).
2. Recipients of multiple solid organ transplants.

| Table 1 | WHO trial registration data set |
|---|---|
| **Data category** | **Information** |
| Primary registry and trial identifying number | ClinicalTrials.gov<br>NCT04941911 |
| Date of registration in primary registry | 28 June 2021 |
| Source(s) of monetary or material support | National Institute of Health Research |
| Primary sponsor | University College London, UK |
| Contact for public queries | MS (michaelspiro@nhs.net) |
| Contact for scientific queries | MS (michaelspiro@nhs.net) |
| Public title | A feasibility study of octreotide infusion during liver transplant |
| Scientific title | A double-blind randomised placebo-controlled feasibility study to assess the impact of octreotide infusion during liver transplantation on post-operative renal failure |
| Countries of recruitment | UK |
| Health condition(s) or problem(s) studied | Liver transplantation |
| Intervention(s) | Active comparator: intravenous octreotide acetate infusion, 100 mcg bolus with a subsequent infusion of 100 mcg per hour during surgery |
| | Placebo comparator: sodium chloride 0.9% w/v |
| Key inclusion and exclusion criteria | Ages eligible for study: ≥18 years<br>Sexes eligible for study: both<br>Accepts healthy volunteers: no |
| | Inclusion criteria: adults aged 18 years and over undergoing primary liver transplantation of a whole or partial liver graft from a cardiac or brain dead donor |
| | Exclusion criteria: previous solid organ transplant, acute liver failure, fulminant hepatic failure, patients receiving a living donor liver graft, patients currently admitted to ICU prior to transplantation, requirement of haemodialysis or continuous veno-venous haemofiltration (CVVHF) preoperatively, known allergy or adverse reaction to octreotide, preoperative decision to use intra-operative CVVHF, a positive pregnancy test. |
| Study type | Interventional |
| | Allocation: patients will be randomised in a 2:1 ratio to either octreotide or placebo groups. Stratified randomisation of patients by source of liver graft (brain death or cardiac death). Masking: quadruple (participant, care provider, investigator, outcomes assessor) |
| | Primary purpose: feasibility |
| | Phase II |
| Date of first enrolment | July 2021 |
| Target sample size | 30 |
| Recruitment status | Pending recruitment |
| Primary outcome(s) | Trial feasibility |
| Key secondary outcomes | Incidence of acute kidney injury<br>Postoperative incidence of a new requirement for renal replacement therapy<br>Incidence of new chronic kidney disease or deterioration of chronic kidney disease<br>Incidence of early allograft dysfunction<br>Patient mortality<br>Intra-operative red blood cell salvage<br>Volume of packed red blood cell transfusion administered<br>Incidence of adverse events secondary to study drug infusion<br>PROMs data |

ICU, intensive care unit; PROMs, patient recorded outcome measures.

3. Previous receipt of a solid organ transplant.
4. Patients with acute or fulminant hepatic failure.
5. Patients receiving a live-donor liver graft.
6. Patients admitted to an intensive care unit (ICU) at the time of transplantation.
7. Preoperative requirement for RRT, including dialysis.
8. Known allergy or adverse reaction to octreotide.
9. Preoperative decision to use intraoperative RRT.
10. Patients enrolled in another interventional study that affects patient care during transplantation surgery or would be expected to have an effect on the study outcomes will be excluded from this study.

Patients who are enrolled in a study examining the impact of extracorporeal machine perfusion of the donated liver prior to transplantation will be eligible for this study.

### Patient identification and recruitment

Participant screening and registration will be undertaken locally at each study site. Overall trial flow is outlined in online supplemental appendix 1.

Patients will be identified and recruited in two ways, depending on whether they are undergoing assessment for suitability for liver transplantation or have already been accepted onto the waiting list for transplantation. Screening will be performed by the clinical team, as will primary contact regarding engagement with research, prior to research team contact.

i. Patients undergoing assessment for liver transplantation will be sent the patient information sheet (PIS) for this study. Patients that meet the inclusion criteria will be approached during their in-patient assessment process to explain the study and for preliminary informed consent.

ii. Patients already listed for liver transplantation who meet the inclusion criteria will be sent the PIS. These patients will be contacted by phone, the study and PIS explained, and preliminary informed consent taken.

Due to the waiting time for transplant and the fixed duration of this study many patients who would be willing to enter the study will not be able to participate because a donor organ does not become available. Both groups of patients will therefore have their consent confirmed on the day of admission for surgery at which point, once the transplant surgery has been confirmed, the patient will be considered to be enrolled.

### Consent

Informed consent will be obtained by a member of the research team and confirmed when the patient is admitted for surgery. The PIS and consent form (online supplemental appendices 2 and 3) will be reviewed and updated, if necessary, throughout the trial (eg, where new safety information becomes available). Consent will be sought for use of participant data in further studies in relation to this research.

### Study interventions

Patients will follow the standard protocol for liver transplantation at each study centre with the only study intervention being the provision of octreotide or placebo infusion. The study infusion will commence after induction of anaesthesia and prior to surgical incision. There will be no cross-over or dose escalation. The study infusate will be administered intravenously through an electronic infusion pump until completion of 50 mL of study medication or transfer to ICU, whichever is sooner. The maximum total dose of octreotide infused will be 1 mg, including an initial slow bolus followed by an infusion. Similarly, the maximum dose of 0.9% saline placebo administrated will be 50 mL. Where the infusion is

completed prior to the end of surgery then no further study drug or placebo will be given.

### Intervention group

Syringes will contain 50 mL of octreotide acetate at 20 mcg/mL in 0.9% w/v sodium chloride in water.

The octreotide infusion will commence with a 100 mcg bolus (5 mL) over 30 min with a subsequent infusion of 100 mcg (5 mL) per hour during surgery.

### Comparator group

Syringes will contain 50 mL of 0.9% w/v sodium chloride in water.

The control infusion will commence with a 5 mL bolus over 30 min with a subsequent infusion of 5 mL per hour during surgery.

### Study drug

Octreotide is licensed in the UK and is being used outside of its marketed authorisation for the purpose of this trial. The Summary of Product Characteristics will be used as a safety reference and the dose will not exceed the maximum dose of 1.5 mg/day. A suitable marketed product will be sourced by the RFH Pharmacy Manufacturing Unit that allows 7 days stability after reconstitution. The study drug will be prepared and released by a Qualified Person for use in the study by the holder of a Manufacturing and Import Authorisation (Investigational Medicinal Products) licence.

### Overdose

Deviations from the treatment schedule or inadvertent administration of additional octreotide may lead to overdose. These will be reported in the case report form (CRF), the patient will be notified and the information and implications reported to the Sponsor. The patient would continue to be included in the trial. Where a serious adverse event (SAE) occurs in association with an overdose then the nature, dose and cause of the overdose will be included in an SAE report form. Participants who receive an overdose of octreotide will be followed up and treated in the usual manner for the trial.

Relevant symptoms of overdose include arrhythmia, hypotension, cardiac arrest, pancreatitis, hepatic steatosis, diarrhoea and lactic acidosis. Management is supportive.

### Randomisation

Consented patients who proceed to transplantation will be randomised. Remote-site computerised random allocation will be performed with a centrally generated code (www.sealedenvelope.com) that will correlate with a coded syringe held in the secure study drug storage facility at each study centre. Patients will be stratified within the randomisation process by source of liver graft (brain death, DBD, or cardiac death donor, DCD) to ensure that patients are balanced for these factors across the two treatment groups. Patients will be randomly allocated to either the octreotide or placebo study groups in a 2:1 ratio. The randomisation list will be held centrally at

a dedicated and independent off-site facility that will be contactable 24 hours a day.

## Allocation concealment

The syringes of octreotide 20 mcg/mL (intervention) and sodium chloride 0.9% w/v (control) will appear identical and only identifiable by their trial-allocated code. No members of the research or clinical team will be aware of patient allocation or syringe contents. The allocation code will only be broken following completion of patient follow-up, data cleaning and analysis; unless an emergency break code is requested. The study statistician will be provided only with information on treatment A or B.

### Unblinding

Where required for medical or safety reasons a trial break code can be obtained from a nominated member of the research team who will remain blinded to patient allocation. Information on the indication and outcome of the unblinding will be forwarded to the Trial Steering Committee (TSC) and included in the final trial report. While Suspected Unexpected Serious Adverse Reaction (SUSAR) reports will include the treatment allocation, this information will not be available to the trial team and blinding will be maintained.

## Outcomes
### Primary outcome

The primary outcomes for this study are based on determining the feasibility of conducting an RCT of octreotide versus placebo infusion during liver transplantation. The feasibility will be assessed by:

► Recruitment ratio (percentage of eligible patients undergoing transplantation who provide consent, target: ≥30%).
► Percentage of eligible patients who receive the study drug infusion in a blinded manner (target: ≥80% of patients).
► Incidence of drug-related serious unexpected adverse events (target: ≤20%).
► Completion of follow-up data collection (target: ≥90% patients).
► Patient refusal to enrol in study on admission for transplantation (target: ≤15%).
► Clinician refusal for inclusion or randomisation following patient recruitment (target: ≤15%).

### Secondary outcomes

A number of secondary outcome measures will be collected to help shape future trials in terms of assessing potential primary outcome measures and monitoring safety end points.

► Intraoperative cell salvage volume (as a proxy for blood loss).
► Intraoperative urine output during each surgical phase.
► Mean arterial blood pressure during each surgical phase.

► Requirement for and mean vasopressor and/or inotrope infusion rate during each surgical phase.
► Volume or mass of de novo packed red blood cells and other clotting products during surgery and on the ICU (within first 24 hours, 72 hours and 1 week).
► Incidence of AKI as defined by the Acute Kidney Injury Network stage 1 criteria (a 50% increase in serum creatinine from baseline or less than 0.5 mL/kg/hours urine output for 6–12 hours post transplant) within 24 hours, 72 hours, 1 week.
► Incidence of postoperative RRT at 24 hours, 72 hours, 1 and 2 weeks postoperatively.
► Incidence of new or worsened CKD at 30 and 90 days.
► Mortality, early allograft dysfunction, graft loss and liver graft function at 30 and 90 days.
► Patient recorded outcome measures (PROM) preoperatively and at 90 days postoperatively;
  – Liver Disease Quality of Life questionnaire.[28]
  – EuroQOL-5D-5L questionnaire.[29]

Major safety outcomes will include:
► Unexpected or resistant bradycardia,.
► Abnormal QTc interval (460 ms in men, 470 ms in women) or associated ventricular arrhythmia or Torsades de Pointes.
► Unexpected or resistant hypoglycaemia (blood sugar <4 mmol/L).
► Potential allergic or anaphylactic reaction to any medication.
► Any report of an abnormal response to codeine or morphine.
► Development of venous or arterial thrombosis.
► Cardiac event including acute coronary syndrome, new heart failure, arrhythmia or resuscitated cardiac arrest.

## Participant withdrawal

A participant may be withdrawn from the trial by their clinician whenever continued participation is considered to no longer be in the participant's best interests, but the reasons for doing so must be recorded. In these cases routine follow-up and safety data will be collected but not any further study-specific data.

A participant may withdraw from the trial at any point without impact on their ongoing care. No further data would be collected. However, data up to that point will be retained. Permission will be sought from patients wishing to withdraw to allow use of their routine follow-up data for trial purposes, especially safety data.

## Sample size

Thirty patients will be recruited from two sites, with 20 in the study drug (octreotide) arm and 10 in the placebo. This sample will allow us to demonstrate the feasibility of conducting a full-scale randomised trial. This study is not powered to detect differences in treatment effect but will obtain estimates of an effect and its variance.[30] This sample size will allow us to estimate patient consent rate with a 95% CI of ± 10% (based on 100 participants

approached). The secondary outcome data will provide the required information for a sample size calculation for the primary end point of a subsequent trial.[31] Survival to discharge is above 96% for both study sites and hence it is likely that the ninety day follow data collection will be achieved for the majority of patients.

## Recruitment rate

Over 275 patients per year will be eligible for study inclusion across the two study sites. With a 10 month recruitment period, this requires a 13% recruitment rate. A recent interventional trial in liver transplantation at the RFH achieved a 53% recruitment rate.[32] Staff involvement and recruitment strategies will be similar in this study indicating this is a deliverable recruitment target.

## Data collection

Data will be collected on trial-specific electronic CRFs (eCRFs) and anonymised data collated on a secure central trial database (MACRO, Elsevier, Amsterdam, Netherlands) on a server in University College London (UCL). Physical trial material will be stored locally at each study site in a secure setting. A trial-specific data management standard operating procedure will contain details of the software to be used for the database, the process of database design, data entry, data quality checks, data queries, data security and database lock. At the end of the trial, all essential documentation will be archived securely for a minimum of 20 years from the declaration of the end of the trial.

All in-hospital data required for the study is part of routine medical care and as such is expected to have a high rate of completeness. Out-of-hospital data is similarly part of routine medical care. The completeness of study-specific PROM questionnaires performed before and after transplantation will be maximised through reminders by telephone and during clinic appointments.

Data will be analysed for completeness and accuracy through a check of 10% of patient records against the source data.

Third party data access will be assessed on a case-by-case basis by the Trial Management Group (TMG), approved by the TSC and limited specific anonymised data will be released, as appropriate. Patients will be consented for further analysis of anonymised data.

## Statistical methods

A statistical analysis plan will be approved by the TSC, prior to data analysis.

The following variables will be used to assess baseline comparability of the randomised groups:
► Study centre at which transplant performed.
► Age.
► Sex.
► The model for end-stage liver disease score.[33]
► The UK end-stage liver disease score.[34]
► Aetiology of liver disease.
► The UK donor risk index.[35]

► Organ graft undergoing normothermic machine perfusion.
► Presence of portal hypertension.
► Severity of preoperative renal dysfunction (estimated glomerular filtration rate).
► American Society of Anesthesiologists score.
► Pr-operative haemoglobin level.
► EuroQol-5D-5L questionnaire scores.
► Liver disease Quality of Life questionnaire score.

Patients will be analysed according to the groups to which there were randomised.

## Primary outcomes

The primary outcome feasibility measures will be quantified using numbers (%). The degree of precision will be quantified using exact 95% CIs.

## Secondary outcomes

Numerical outcomes will be summarised using either mean (SD) or medians (IQR) as appropriate. Binary and ordinal outcomes will be summarised using number (%). The degree of precision will be quantified using 95% CIs. Differences between groups will be quantified using differences in means or proportions, as appropriate with corresponding 95% CIs.

Since this is a feasibility study, no formal statistical tests are planned.

## Sensitivity and other planned analyses

No sensitivity analyses are planned. There will be no interim analysis.

We will perform exploratory subgroup analyses for donor organ type (DCD and DBD), the presence of CKD and known portal thrombosis (or if diagnosed at transplantation). We will also assess the impact of surgical approach (cross-clamp vs any other approach), the use of machine perfusion vs cold storage and hospital site.

## Patient replacement

Patients who are randomised and allocated to a treatment arm but subsequently do not complete the trial will not be replaced.

## Monitoring

### Oversight committees

A TMG and TSC have been formed. The terms of reference and functions of these committees are provided in online supplemental appendix 4.

The TMG will include the Chief Investigators, coinvestigators, trial staff, patient expert and study site principle investigators. The TMG will be responsible for overseeing the trial. The TMG will review recruitment figures, SAEs and substantial amendments to the protocol prior to submission to the Research Ethics Committee (REC).

### Data monitoring

The TSC will represent a combined TSC and Data Monitoring Committee with expertise in performing and reporting clinical trials, statistical methods and clinical

expertise for evaluation of post-transplant complications. The role of the TSC is to provide oversight for the trial and provide advice through its independent Chair to the Chief Investigator, the TMG, trial Sponsor, Funder, and host institutions on all aspects of the trial. The TSC will recommend any appropriate amendments/actions for the trial as necessary. The TSC is independent of the TMG, Funder and Sponsor and members have declared no competing interests.

### Early stopping guidelines

The trial may be stopped before completion either on the recommendation of the TSC or the sponsor and chief investigator. There will be no statistical stopping rules.

### Adverse events

All adverse events will be recorded in the medical records in the first instance. All SAEs within 90 days of patient enrolment will be recorded in the eCRF with clinical symptoms, a brief description and date of the event. SAEs will be reported to the Sponsor within 24 hours. Where the event is unexpected and thought to be related to the intervention, this will be reported to the Health Research Authority within 15 days. Participants suffering an SAE will be followed up until clinical recovery is complete and laboratory results have returned to expected values, or until the event has stabilised. Follow-up will continue after completion of trial follow-up if necessary. For the purpose of analysis, SAEs prior to discharge, within 30 days and within 90 days of enrolment will be analysed and published.

The sponsor will notify the REC and Medicines and Healthcare products Regulatory Agency (MHRA) of all SUSARs within 15 days. SUSARs that are fatal or life threatening will be notified to the MHRA and REC within 7 days of the sponsor learning of them.

### Auditing

Given the short duration of this study no formal audit is planned. All documentation will be collected and recorded in keeping with Good Clinical Practice guidelines and will be available for external review or auditing if required.

## ETHICS AND DISSEMINATION
### Research ethics approval

This study has been given approval by the East Midlands—Leicester South REC (Integrated Research Application System reference: 278918, REC reference: 21/EM/0076) and the Health Research Authority. This feasibility study does not require MHRA approval.

### Protocol amendments

Substantial protocol amendments will be reviewed by the TMG, TSC and sponsor prior to REC and Health Research Authority submission.

Any changes from this protocol paper will be highlighted in the study outcome publication.

### Ancillary and post-trial care

The sponsor holds insurance against claims from participants for injury caused by their participation in the trial. Each study site hospital will provide negligence insurance cover for harm caused by their employees. All ongoing care for the treatment of adverse events will be provided by the recruiting hospital.

### Dissemination policy

A publication policy will be written and agreed by the TMG prior to trial data analysis. This will be agreed with the TSC. The main study outcomes will be submitted to an open-access journal.

### Patient and public involvement

Interviews and engagement with liver transplantation patients and their families has been a key motivator behind this study and a source of knowledge to identify outcomes that have been actively promoted in study design. A reduction in perioperative complications, especially haemorrhage, was identified by patients as a main research priority as well as physical and psychosocial well-being and quality of life after surgery. The PROM questionnaires were assessed during patient and family interviews to ascertain their accessibility, value and feasibility. The PIS has been reviewed with preoperative patients and family to obtain feedback on its clarity, readability and content. A transplant recipient patient expert is a coapplicant to this grant and will be an active member of the TMG. Our patient expert will assist us in providing effective communication with patients and public with a patient-centred viewpoint.

**Contributors** JF, GA, NRW, DM, BRD and MS: grant applicant, drafting, editing and approving this manuscript. BS: trial manager, editing and approving this manuscript.

**Funding** This project is funded by the National Institute for Health Research (NIHR) under its Research for Patient Benefit (RfPB) Programme (Grant Reference Number PB-PG-0817-20023).

**Disclaimer** The views expressed are those of the author(s) and not necessarily those of the NIHR or the Department of Health and Social Care.

**Competing interests** None declared.

**Patient consent for publication** Not applicable.

**Provenance and peer review** Not commissioned; externally peer reviewed.

**ORCID iD**
Jeremy Fabes http://orcid.org/0000-0003-1111-5973

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
