## [Reviewer comments · BMJ Open]

ARTICLE DETAILS

TITLE (PROVISIONAL)	Protocol for a prospective double-blind, randomised, placebo-controlled feasibility trial of octreotide infusion during liver transplantation
AUTHORS	Fabes, Jez; Ambler, Gareth; Shah, Bina; Williams, Norman; Martin, Daniel; Davidson, Brian; Spiro, Michael

VERSION 1 – REVIEW

REVIEWER	Gavriilidis, Paschalis Queen Elizabeth Hospital Birmingham
REVIEW RETURNED	02-Aug-2021

GENERAL COMMENTS	It is well scheduled study based on sound methodology.
--

REVIEWER	Otero, Alejandra A Coruña University Hospital, Unidad de Trasplante hepático
REVIEW RETURNED	09-Aug-2021

GENERAL COMMENTS	Congratulations for the job and waiting for the results.
--

REVIEWER	Zheng, Shusen First Affiliated Hospital, School of Medicine, Zhejiang University, Division of Hepatobiliary and Pancreatic Surgery, Department of Surgery
REVIEW RETURNED	10-Aug-2021

GENERAL COMMENTS	Liver transplantation (LT) is often associated with major hemorrhage and a large red blood cell (RBC) transfusion requirement, which lead to increased incidence of acute kidney injury and even associated with increased morbidity and mortality of patients. This study was designed to investigate the protective effect of octreotide infusion by recruiting patients to a double-blind, randomised, placebo-controlled trial of intravenous octreotide during LT, and may help elucidating the effect of octreotide on RBC transfusion and renal replacement therapy requirements. In addition, this prospective study aims to explore the harmful effects of octreotide and therefore may help clinicians to make better decisions on octreotide administration during LT. However, there are two concerns in the text require further consideration. 1. LT is the most rational therapeutic option for patients with different kinds of liver diseases, for example, hepatocellular carcinoma (HCC). Currently, LT for HCC represents 15–50% of all liver transplants performed in most centres. In the exclusion criteria, the authors excluded patients with acute or fulminant hepatic failure, while the effect of HCC on the outcome of LT
---

	should be taken into consideration. Whether HCC is associated with hemorrhage risk and therefore affect the result of this study. 2. The authors plan to recruit 30 adult patients at two liver transplant centres, it's crucial to exclude multicenter confounding factors such as perioperative management and surgical practice.
--	--

VERSION 1 – AUTHOR RESPONSE

Many thanks for your invitation to provide revisions to our manuscript, we have made these changes and our responses are included here.

We have included the feasibility outcomes in the abstract methods.

In response to Reviewer 3;

Many thanks for your review and comments on the protocol manuscript.

In a recent analysis of our transplant population at the Royal Free Hospital, less than 1% of patients were undergoing transplantation for acute or acute-on-chronic liver failure. We have excluded this cohort as it is not possible to perform the necessary phases of the study such as recruitment, provision of proper consent and data collection such as baseline quality of life questionnaires. We anticipate that HCC patients will constitute a large proportion of our study population, so we expect our results to be highly relevant to this group and any associated haemorrhage risk.

Regarding your second point, we fully recognise the importance of balanced patient cohorts to minimise confounding factors such as differences between recruiting centres. However, as this is a feasibility study, we do not intend to make any comments on the efficacy of octreotide, rather to demonstrate that the study design is practical. During the main, substantive, trial that follows this feasibility study we will have appropriate stratification of our allocation randomisation, including by transplant centre, to fully address this source of bias.

VERSION 2 – REVIEW

REVIEWER	Zheng, Shusen First Affiliated Hospital, School of Medicine, Zhejiang University, Division of Hepatobiliary and Pancreatic Surgery, Department of Surgery
REVIEW RETURNED	03-Nov-2021
GENERAL COMMENTS	The topic of this study is relevant to the clinical practice in liver transplantation .This Protocol was well orgnized and written, especially welcomed to tranplantation doctors. The authors addressed the limitations of this study as well.